# ON THE DYNAMIC REGRET OF ONLINE MULTIPLE MIRROR DESCENT

## ABSTRACT

We study the problem of online convex optimization, where a learner makes sequential decisions to minimize an accumulation of strongly convex costs over time. The quality of decisions is given in terms of the dynamic regret, which measures the performance of the learner relative to a sequence of dynamic minimizers. Prior works on gradient descent and mirror descent have shown that the dynamic regret can be upper bounded using the path length, which depend on the differences between successive minimizers, and an upper bound using the squared path length has also been shown when multiple gradient queries are allowed per round. However, they all require the cost functions to be Lipschitz continuous, which imposes a strong requirement especially when the cost functions are also strongly convex. In this work, we consider Online Multiple Mirror Descent (OMMD), which is based on mirror descent but uses multiple mirror descent steps per online round. Without requiring the cost functions to be Lipschitz continuous, we derive two upper bounds on the dynamic regret based on the path length and squared path length. We further derive a third upper bound that relies on the gradient of cost functions, which can be much smaller than the path length or squared path length, especially when the cost functions are smooth but fluctuate over time. Thus, we show that the dynamic regret of OMMD scales linearly with the minimum among the path length, squared path length, and sum squared gradients. Our experimental results further show substantial improvement on the dynamic regret compared with existing alternatives.

## 1 INTRODUCTION

Online optimization refers to the design of sequential decisions where system parameters and cost functions vary with time. It has applications to various classes of problems, such as object tracking (Shahrampour & Jadbabaie, 2017), networking (Shi et al., 2018), cloud computing (Lin et al., 2012), and classification (Crammer et al., 2006). It is also an important tool in the development of algorithms for reinforcement learning (Yuan & Lamperski, 2017) and deep learning (Mnih et al., 2015).

In this work, we consider online convex optimization, which can be formulated as a discrete-time sequential learning process as follows. At each round $t$, the learner first makes a decision $x_t \in \mathcal{X}$, where $\mathcal{X}$ is a convex set representing the solution space. The learner then receives a convex cost function $f_t(x) : \mathcal{X} \to \mathbb{R}$ and suffers the corresponding cost of $f_t(x_t)$ associated with the submitted decision. The goal of the online learner is to minimize the total accrued cost over a finite number of rounds, denoted by $T$. For performance evaluation, prior studies on online learning often focus on the *static* regret, defined as the difference between the learner's accumulated cost and that of an optimal fixed offline decision, which is made in hindsight with knowledge of $f_t(\cdot)$ for all $t$:

$$\text{Reg}_T^s = \sum_{t=1}^T f_t(x_t) - \min_{x \in \mathcal{X}} \sum_{t=1}^T f_t(x).$$

A successful online algorithm closes the gap between the online decisions and the offline counterpart when normalized by $T$, i.e., sustaining sublinear static regret in $T$. In the literature, there are various online algorithms (Zinkevich, 2003; Cesa-Bianchi & Lugosi, 2006; Hazan et al., 2006; Duchi et al., 2010; Shalev-Shwartz, 2012) that guarantee a sublinear bound on the static regret.

However, algorithms that guarantee performance close to that of a static decision may still perform poorly in dynamic settings. Consequently, the static regret fails to accurately reflect the quality of decisions in many practical scenarios. Therefore, the *dynamic* regret has become a popular metric in recent works (Besbes et al., 2015; Mokhtari et al., 2016; Yang et al., 2016; Zhang et al., 2017), which allows a dynamic sequence of comparison targets and is defined by

$$\text{Reg}_T^d = \sum_{t=1}^{T} f_t(x_t) - \sum_{t=1}^{T} f_t(x_t^*),$$

where $x_t^* = \text{argmin}_{x \in \mathcal{X}} f_t(x)$ is a minimizer of the cost at round $t$.

It is well-known that the online optimization problem may be intractable in a dynamic setting, due to arbitrary fluctuation in the cost functions. Hence, achieving a sublinear bound on the dynamic regret may be impossible. However, it is possible to upper bound the dynamic regret in terms of certain regularity measures. One of the measures to represent regularity is the *path length*, defined by

$$C_T = \sum_{t=2}^{T} \|x_t^* - x_{t-1}^*\|, \tag{1}$$

which illustrates the accumulative variation in the minimizer sequence. For instance, the dynamic regret of online gradient descent for convex cost functions can be bounded by $O(\sqrt{T}(1 + C_T))$ (Zinkevich, 2003). [1] For strongly convex functions, the dynamic regret of online gradient descent can be reduced to $O(C_T)$ (Mokhtari et al., 2016). When the cost functions are smooth and strongly convex, by allowing the learner to make multiple queries to the gradient of the cost functions, the regret bound can be further improved to $O(\min(C_T, S_T))$, where $S_T$ represents the *squared path length*, defined by

$$S_T = \sum_{t=2}^{T} \|x_t^* - x_{t-1}^*\|^2, \tag{2}$$

which can be smaller than the path length when the distance between successive minimizers is small. All the aforementioned studies require the cost functions to be Lipschitz continuous. However, there are many commonly used cost functions, e.g., the quadratic function that do not meet the Lipschitz condition. In addition, the above works rely on measuring distances using Euclidean norms, which hinders the projection step in gradient descent update for some constraint sets, e.g., probability simplex (Duchi, 2018).

Besides gradient descent, mirror descent is another well-known technique of online convex optimization (Hall & Willett, 2015; Jadbabaie et al., 2015). Mirror descent uses the Bregman divergence, which generalizes the Euclidean norm used in the projection step of gradient descent, thus acquiring expanded applicability to a broader range of problems. In addition, the Bregman divergence is only mildly dependent on the dimension of decision variables (Beck & Teboulle, 2003; Nemirovsky & Yudin, 1983), so that mirror descent is optimal among first-order methods when the decision variables have high dimensions (Duchi et al., 2010). In this work we focus on the mirror descent approach.

In previous works on online mirror descent, the learner queries the gradient of each cost function only once, and performs one step of mirror descent to update its decision (Hall & Willett, 2015; Shahrampour & Jadbabaie, 2017). In this case, the dynamic regret has an upper bound of order $O(\sqrt{T}(1 + C_T))$, which is the same as that of online gradient descent in (Zinkevich, 2003). In this work, we investigate whether it is possible to improve the dynamic regret when the learner performs multiple mirror descent steps in each online round, while relaxing the Lipschitz continuity condition on the cost functions.

To this end, we analyze the performance of the Online Multiple Mirror Descent (OMMD) algorithm, which uses multiple steps of mirror descent per online round. When the cost functions are smooth and strongly convex, we show that the upper bound on the dynamic regret can be reduced from

---

[1] A more general definition of the dynamic regret was introduced in (Zinkevich, 2003), which allows comparison against an arbitrary sequence $\{u_t\}_{t=1}^{T}$. We note that the regret bounds developed in (Zinkevich, 2003) also hold for the specific case of $u_t = x_t^*$.

$O(\sqrt{T}(1 + C_T))$ to $O(\min(C_T, S_T, G_T))$, where $G_T$ represent the *sum squared gradients*, i.e.,

$$G_T = \sum_{t=1}^{T} \|\nabla f_t(x_t)\|_*^2, \tag{3}$$

where $\|.\|_*$ denotes the dual norm. The sum squared gradients $G_T$ can be smaller than both the path length and squared path length, especially when the cost functions fluctuate drastically over time. In contrast to the aforementioned works, our analysis does not require the cost functions to be Lipschitz continuous. Furthermore, our numerical experiments suggest substantially reduced dynamic regret compared with the best known alternatives, including single-step dynamic mirror descent (Hall & Willett, 2015), online multiple gradient descent (Zhang et al., 2017), and online gradient descent Zinkevich (2003).

## 2 ONLINE MULTIPLE MIRROR DESCENT

In this section, we describe OMMD and discuss how the learner can improve the dynamic regret by performing multiple mirror descent steps per round. Before delving into the details, we proceed by stating several definitions and standard assumptions.

### 2.1 PRELIMINARIES

*Definition 1*: The Bregman divergence with respect to the regularization function $r(\cdot)$ is defined as
$$D_r(x, y) = r(x) - r(y) - \langle \nabla r(y), x - y \rangle.$$

The Bregman divergence is a general distance-measuring function, which contains the Euclidean norm and the Kullback-Leibler divergence as two special cases.

Using the Bregman divergence, a generalized definition of strong convexity is given in (Shalev-Shwartz & Singer, 2007).

*Definition 2*: A convex function $f(\cdot)$ is $\lambda$-strongly convex with respect to a convex and differentiable function $r(\cdot)$ if
$$f(y) + \langle \nabla f(y), x - y \rangle + \lambda D_r(x, y) \le f(x), \ \forall x, y \in \mathcal{X}.$$

Following many prior studies on mirror descent, we assume that the cost functions are $\lambda$-strongly convex, where the above generalized strong convexity definition is used. We further assume that the cost functions are $L$-smooth, and the regularization function $r(\cdot)$ is $L_r$-smooth and 1-strongly convex with respect to some norm (refer to App. A for definitions). We note that these are standard assumptions commonly used in the literature after the group of studies began by (Hazan et al., 2006; Shalev-Shwartz & Singer, 2007), to provide stronger regret bounds by constraining the curvature of cost functions.

We further make a standard assumption that the Bregman divergence is Lipschitz continuous as follows:
$$|D_r(x, z) - D_r(y, z)| \le K\|x - y\|, \ \forall x, y, z \in \mathcal{X},$$
where $K$ is a positive constant. We note that this condition is much milder than the condition of Lipschitz continuous *cost functions* required in (Zhang et al., 2017; Mokhtari et al., 2016; Hall & Willett, 2015). There is a notable weakness in such bounds. Since the sequence of cost functions are revealed to the learner, the learner has no control over it. If these cost functions happen to not meet the Lipschitz condition, earlier analyses that require this condition become inapplicable. In this work, we do not require the cost functions to be Lipschitz continuous. Instead, we move the Lipschitz continuity condition from the cost functions to the Bregman divergence to broaden the application of our work. The main benefit of this is that the regularization function and the corresponding Bregman divergence is within the control of the learner. The learner can carefully design this regularization function to satisfy the Lipschitz continuity of the associated Bregman divergence with a small factor. For example, in the particular case of the KL divergence, which is obtained by the choosing negative entropy as the regularization function, on the set $\mathcal{X} = \{x | \sum_{i=1}^{d} x_i = 1; x_i \ge \frac{1}{D}\}$, the constant $K$ is of $O(\log D)$. Other examples of many widely used Bregman divergences that satisfy this condition are given in (Bauschke & Borwein, 2001).

---

**Algorithm 1** Online Multiple Mirror Descent

---

**Input:** Arbitrary initialization of $x_1 \in \mathcal{X}$; step size $\alpha$; time horizon $T$.
**Output:** Sequence of decisions $\{x_t : 1 \le t \le T\}$.
1: **for** $t = 1, 2, \ldots, T$ **do**
2:     submit $x_t \in \mathcal{X}$ and receive $f_t(\cdot)$
3:     set $y_t^1 = x_t$
4:     **for** $i = 1, 2, \ldots, M$ **do**
5:        $y_t^{i+1} = \text{argmin}_{y \in \mathcal{X}} \{ \langle \nabla f_t(y_t^i), y \rangle + \frac{1}{\alpha} D_r(y, y_t^i) \}$
6:     **end for**
7:     set $x_{t+1} = y_t^{M+1}$
8: **end for**

---

## 2.2 ONLINE CONVEX OPTIMIZATION WITH OMMD

We consider online optimization over a finite number of rounds, denoted by $T$. At the beginning of every round $t$, the learner submits a decision represented by $x_t$, which is taken from a convex and compact set $\mathcal{X}$. Then, an adversary selects a function $f_t(\cdot)$ and the learner suffers the corresponding cost $f_t(x_t)$. The learner then updates its decision in the next round. With standard mirror descent, this is given by

$$x_{t+1} = \underset{x \in \mathcal{X}}{\text{argmin}} \{ \langle \nabla f_t(x_t), x \rangle + \frac{1}{\alpha} D_r(x, x_t) \} \tag{4}$$

where $\alpha$ is a fixed step size, and $D_r(\cdot, \cdot)$ is the Bregman divergence corresponding to the regularization function $r(\cdot)$. The update in equation 4 suggests that the learner aims to stay close to the current decision $x_t$ as measured by the Bregman divergence, while taking a step in a direction close to the negative gradient to reduce the current cost at round $t$.

OMMD uses mirror descent in its core as the optimization workhorse. However, in contrast to classical online optimization methods, where the learner queries the gradient of each cost function only once, OMMD is designed to take advantage of the curvature of cost functions by allowing the learner to make multiple queries to the gradient in each round. This is especially important when the successive cost functions have similar curvatures. In particular, in order to track $x_{t+1}^*$ the learner needs to access the gradient of the cost function, i.e., $\nabla f_{t+1}(\cdot)$. Unfortunately, this information is not available until the end of round $t + 1$. However, if the successive functions have similar curvatures, the gradient of $f_t(\cdot)$ is a reasonably accurate estimate for the gradient of $f_{t+1}(\cdot)$. In this case, every time that the learner queries the gradient of $f_t(\cdot)$, it finds a point that is likely to be closer to the minimizer of $f_{t+1}(\cdot)$. Hence, it may benefit the learner to perform multiple mirror descent steps in each round.

Thus, the learner generates a series of decisions, represented by $y_t^1, y_t^2, \ldots, y_t^{M+1}$, via the following updates:

$$y_t^1 = x_t, \quad y_t^{i+1} = \underset{y \in \mathcal{X}}{\text{argmin}} \{ \langle \nabla f_t(y_t^i), y \rangle + \frac{1}{\alpha} D_r(y, y_t^i) \}, \quad i = 1, 2, \ldots, M. \tag{5}$$

Then, by setting $x_{t+1} = y_t^{M+1}$, the learner proceeds to the next round, and the procedure continues. Note that $M$ is independent of $T$.

Applying multiple steps of mirror descent can reveal more information about the sequence of minimizers. It can reduce the dynamic regret, but only if the series of decisions in equation 5 helps decrease the distance to the minimizer $x_{t+1}^*$. Therefore, quantifying the benefit of OMMD over standard mirror descent requires careful analysis on the impact of the fluctuation of $f_t(\cdot)$ over time. To this end, we provide an analysis to bound the dynamic regret of OMMD in the next section.

## 3 THEORETICAL RESULTS

The following lemma paves the way for the proposed analysis on the dynamic regret of OMMD. It bounds the distance of the learner's future decision from the current optimal solution, after a single step of mirror descent.

**Lemma 1** *Assume that $f_t(\cdot)$ is $\lambda$-strongly convex with respect to a differentiable function $r(\cdot)$, and is L-smooth. Single-step mirror descent with a fixed step size $\alpha \leq \frac{1}{L}$ guarantees the following:*

$$D_r(x_t^*, x_{t+1}) \leq \beta D_r(x_t^*, x_t),$$

*where $x_t^*$ is the unique minimizer of $f_t(\cdot)$, and $\beta = 1 - \frac{2\alpha\lambda}{1+\alpha\lambda}$.*

Lemma 1 is proved in App. B in the supplementary material.

*Remark 1.* Lemma 1 states that a mirror descent step reduces the distance (measured by the Bregman divergence) of the learner's decisions to the current minimizer. This generalizes the results in (Mokhtari et al., 2016; Zhang et al., 2017), where similar bounds were derived for online gradient descent when the distance was measured in Euclidean norms. In particular, those results correspond to the special choice of $r(x) = \|x\|_2^2$, which reduces the Bregman divergence to Euclidean distance, i.e., $D_r(x, y) = \|x - y\|^2$.

Lemma 1 indicates that the distance between the next decision $x_{t+1}$ and the minimizer $x_t^*$ is strictly smaller than the distance between the current decision $x_t$ and the minimizer at round $t$. This implies that if the minimizers of the functions $f_t(\cdot)$ and $f_{t+1}(\cdot)$, which are $x_t^*$ and $x_{t+1}^*$ respectively, are not far from each other, applying mirror descent multiple times enables the online learner to more accurately track the sequence of optimal solutions $x_t^*$.

The succeeding theorems provide three separate upper bounds on the dynamic regret of OMMD, based on path length $C_T$ (as defined in equation 1), squared path length $S_T$ (as defined in equation 2), and sum squared gradients (as defined in equation 3).

**Theorem 2** *Assume that $r(\cdot)$ is $L_r$-smooth and 1-strongly convex with respect to some norm $\|\cdot\|$ and that the cost functions are L-smooth and $\lambda$-strongly convex with respect to $r(\cdot)$. Let $x_t$ be the sequence of decisions generated by OMMD with a fixed step size $\frac{1}{2\lambda} < \alpha \leq \frac{1}{L}$ and $M \geq \lceil (\frac{1}{2} + \frac{1}{2\alpha\lambda}) \log L_r \rceil$ mirror descent steps per round. The dynamic regret satisfies the following bound:*

$$\sum_{t=1}^{T} f_t(x_t) - f_t(x_t^*) \leq \left( \frac{K\lambda}{2\alpha\lambda - 1} \frac{(1 + \sqrt{L_r\beta^M})}{(1 - \sqrt{L_r\beta^M})} \right) (C_T + \|x_1^* - x_1\|).$$

*where $\beta$ is the shrinking factor derived in Lemma 1, and $K$ is the Lipschitz constant associated with $D_r(\cdot, \cdot)$.*

The proof of Theorem 2 is given in App. C in the supplementary material.

*Remark 2.* It has been shown in (Hall & Willett, 2015) that single-step mirror descent guarantees an upper bound of $O(\sqrt{T}(1 + C_T))$ on the dynamic regret for convex cost functions. With that bound, a sublinear path length is not sufficient to guarantee sublinear dynamic regret. In contrast, Theorem 2 implies that OMMD reduces the upper bound to $O(C_T)$ when the cost functions are strongly convex and smooth, which implies that a sublinear path length is sufficient to yield sublinear dynamic regret.

*Remark 3.* The range of $M$ where the bound in Theorem 2 holds is usually wide. For example, it is $M \geq 3$ and $M \geq 5$ for the two experiments shown in Section 4.

**Theorem 3** *Under the same convexity and smoothness conditions stated in Theorem 2, let $x_t$ be the sequence of decisions generated by OMMD with a fixed step size $\alpha \leq \frac{1}{L}$ and $M \geq \lceil (\frac{1}{2} + \frac{1}{2\alpha\lambda}) \log 2L_r \rceil$ mirror descent steps per round. For any arbitrary positive constant $\theta$, the dynamic regret is upper bounded by*

$$\sum_{t=1}^{T} f_t(x_t) - f_t(x_t^*) \leq \sum_{t=1}^{T} \frac{\|\nabla f_t(x_t^*)\|_*^2}{2\theta} + \left( \frac{LL_r + \theta}{1 - 2L_r\beta^M} \right) \left( S_T + \frac{\|x_1^* - x_1\|^2}{2} \right).$$

Theorem 3 is proved in App. D in the supplementary material.

Since the gradient at $x_t^*$ is zero if $x_t^*$ is in the relative interior of the feasibility set $\mathcal{X}$, i.e., $\|\nabla f_t(x_t^*)\| = 0$, the above theorem can be simplified to the following corollary.

**Corollary 4** *If $x_t^*$ belongs to the relative interior of the feasibility set $\mathcal{X}$ for all t, the dynamic regret bound in Theorem 3 is of order $O(S_T)$.*

When the cost functions drift slowly, the distances between successive minimizers are small. Hence, the squared path length $S_T$, which relies on the square of those distances, can be significantly smaller than the path length $C_T$. In this case, Theorem 3 and Corollary 4 can provide a tighter regret bound than Theorem 2.

**Theorem 5** *Under the same convexity and smoothness conditions stated in Theorem 2, let $x_t$ be the sequence of decisions generated by OMMD with a fixed step size $\alpha > \frac{1}{2\lambda}$. The following bound holds on the dynamic regret:*

$$\sum_{t=1}^{T} f_t(x_t) - f_t(x_t^*) \leq \frac{\alpha^2 \lambda}{4\alpha\lambda - 2} G_T.$$

The proof of Theorem 5 is given in App. E in the supplementary material.

*Remark 4.* Interestingly, Theorem 5 implies that sublinear dynamic regret can be achieved when the gradient of the cost functions shrink over time. For instance, if $\|\nabla f_t(x)\|_* = O(1/t^\gamma)$ for some $\gamma > 0$, Theorem 5 guarantees $O(T^{1-2\gamma})$ dynamic regret. This is especially important when the cost functions decrease while the minimizers fluctuate. In this scenario, the path length $C_T$ and squared path length $S_T$ may grow linearly, whereas diminishing gradients ensure sublinear $G_T$.

Theorem 2, Corollary 4, and Theorem 5, respectively, state that the dynamic regret of OMMD is upper bounded *linearly* by path length $C_T$, squared path length $S_T$, and sum squared gradients $G_T$. This immediately leads to the following result.

**Corollary 6** *Under the same convexity and smoothness conditions stated in Theorem 2, the dynamic regret of OMMD with suitably chosen $\alpha$ and $M$ has an upper bound of $O(\min(C_T, S_T, G_T))$.*

*Remark 5.* We note that (Mokhtari et al., 2016) and (Zhang et al., 2017) provide upper bounds of $O(C_T)$ and $O(\min(C_T, S_T))$, respectively, on the dynamic regret of online gradient descent with single and multiple gradient queries, while (Hall & Willett, 2015) presents an upper bound of $O(\sqrt{T}(1 + C_T))$ on the dynamic regret of online single-step mirror descent. Corollary 6 shows that OMMD can improve the dynamic regret bound to $O(\min(C_T, S_T, G_T))$. Furthermore, in contrast to these studies, our analysis does not require the cost functions to be Lipschitz continuous.

The quantities $C_t$, $S_T$, and $G_T$ represent distinct aspects of an online learning problem and are not generally comparable. The following example demonstrates the benefit of having multiple upper bounds and taking their minimum. Consider a sequence of quadratic programming problems of the form $f_t(x) = \|A_t x - b_t\|^2$ over the $d$-dimensional probability simplex. Assume that for any $t \geq 1$, we have the parameter sequence of

$$A_t = \left\{ \begin{array}{ll} \text{diag}(\frac{1}{t^{p_1}}, 0, 0, \ldots, 0), & \text{if } t \text{ is odd} \\ \text{diag}(0, \frac{1}{t^{p_1}}, 0, \ldots, 0), & \text{if } t \text{ is even,} \end{array} \right.$$

and $b = [\frac{1}{t^{p_2}}, \frac{1}{t^{p_2}}, \ldots, \frac{1}{t^{p_2}}]'$, where $p_1$ and $p_2$ are positive constants such that $p_2 \leq p_1$. In this setting, we observe that $C_T = O(T)$ and $G_T = O(T^{1-p_1-p_2})$. Thus, $G_T$ can be considerably smaller than $C_T$. On the other hand, it is also possible that $C_T$ is smaller than $G_T$ in other cases. For example, let $A_t = \text{diag}(1/2, 0, \ldots, 0)$ on odd rounds and $A_t = \text{diag}(0, -1/2, \ldots, -1/2)$ , and $b_t$ be the unity vector for all $t$. In this case, we observe that $C_T = O(1)$, while the sum gradient scales linearly with time, i.e., $G_T = O(T)$. Thus, neither $C_T$ nor $G_T$ alone can provide a small regret bound for all cases. Similar examples can be found in comparison between $S_T$ and $G_T$ but are omitted for brevity.

## 4 EXPERIMENTS

We investigate the performance of OMMD via numerical experiments in two different learning scenarios (with further experiments presented in App. G in the supplementary material). First, we consider a ridge regression problem on the CIFAR-10 dataset (Krizhevsky, 2009). Then, we study a case of online convex optimization where the difference between successive minimizers diminishes as time progresses. We compare OMMD with the following alternatives: Online Gradient Descent (OGD) (Zinkevich, 2003), Online Multiple Gradient Descent (OMGD) (Zhang et al., 2017), and Dynamic Mirror Descent (DMD) (Hall & Willett, 2015).

In the first experiment, we consider multi-class classification with ridge regression. In this task, the learner observes a sequence of labeled examples $(\omega, z)$, where $\omega \in \mathbb{R}^d$, and the label $z$, denoting the class of the data example, is drawn from a discrete space $\mathcal{Z} = \{1, 2, \ldots, c\}$. We use the CIFAR-10 image dataset, which contains $5 \times 10^4$ data samples. Each data sample $\omega$ is a color image of size $32 \times 32$ pixel that can be represented by a 3072-dimensional vector, i.e., $d = 3072$. Data samples correspond to color images of objects, including airplanes, cars, birds, cats, deer, dogs, frogs, horses, ships, and trucks. Hence, there are $c = 10$ different classes. For ridge regression, the cost function associated with batch of data samples at round $t$, i.e., $(\omega_{1,t}, z_{1,t}), \ldots, (\omega_{b,t}, z_{b,t})$, is given by

$$f(x, (\omega_t, z_t)) = \|\omega_t^T x - z_t\|_2^2,$$

where $x$ is the optimization variable, which is constrained by the set $\mathcal{X} = \{x : x \in \mathbb{R}_+^d, \| x \|_1 = 1\}$, and $(\omega_t, z_t)$ compactly represents the batch of data samples at round $t$, i.e., $\omega_t = [\omega_{1,t}, \omega_{2,t}, \ldots, \omega_{b,t}]^T$ and $z_t = [z_{1,t}, z_{2,t}, \ldots, z_{b,t}]^T$. The goal of the learner is to classify streaming images online by tracking the unknown optimal parameter $x_t^*$. We use the negative entropy regularization function, i.e., $r(x) = \sum_{j=1}^d x_j \log(x_j)$, which is strongly convex with respect to the $l1$-norm. Then, the mirror descent update in equation 4 leads to the following update:

$$y_{t,j}^{i+1} = \frac{y_{t,j}^i \exp(-\alpha \nabla f_t(y_{t,j}^i))}{\sum_{j=1}^d y_{t,j}^i \exp(-\alpha \nabla f_t(y_{t,j}^i))}, \tag{6}$$

where $x_{t,j}$ and $\nabla f_t(x_t)_j$ denote the $j$-th component of $x_t$ and $\nabla f_t(x_t)$ respectively. The proof of the above closed-form update is given in App. F in the supplementary material. In our experiment, we set batch size to 20 data samples per online round, and set $\alpha = 0.1$.

In Fig. 1, we compare the performance of OMMD with DMD, OGD, and OMGD in terms of the dynamic regret. We see that the methods based on mirror descent perform better than those based on gradient descent as generally expected. Furthermore, OMMD with $M = 10$ can reduce the dynamic regret up to $30\%$ in comparison with DMD. The dynamic regret associated with all algorithms grow linearly with the number of rounds. This is because the sequence of minimizers $x_t^*$ depend on batches of samples that are independent over time, so that they do not converge. We note that this is common for online optimization in dynamic settings where steady fluctuation in the environment results in linear dynamic regret.

Next, we study the performance of OMMD in solving a sequence of quadratic programming problems of the form $f_t(x_1, x_2) = \rho \|x_1 - a_t\|^2 + \|x_2 - b_t\|^2$, where $\rho$ is a positive constant, $a_t$ and $b_t$ are time-variant vectors, and the decision variable are $x_1 \in \mathbb{R}^{d_1}$, $x_2 \in \mathbb{R}^{d_2}$, such that $d_1 + d_2 = d$. In our experiment, we set $\rho = 10$, $d_1 = 500$, and $d = 1000$. We assume that $b_t$ is time-invariant and for all rounds $t$ we have $b_t = 2$, while $a_t$ satisfies the recursive formula $a_{t+1} = a_t + 1/\sqrt{t}$ with initial value $a_1 = -1.5$. We further set the step size $\alpha = 0.03$. We use the same regularization function and constraint set as in the previous experiment.

From Fig. 2, we observe that the performance advantage of OMMD is even more pronounced. As time progresses and the difference between the successive cost functions becomes less significant, the difference between the minimizers decreases. In this case, OMMD can significantly improve the performance of online optimization by reducing the gap between the learner's decisions and the minimizers sequence. In particular, compared with DMD, OMMD with $M = 10$ reduces the dynamic regret up to $80\%$ after 2500 rounds.

## 5 RELATED WORKS

The problem of online convex optimization has been extensively studied in the literature since the seminal work of (Zinkevich, 2003). Most prior works study various online algorithms that guarantee a sublinear bound on the static regret (Zinkevich, 2003; Cesa-Bianchi & Lugosi, 2006; Hazan et al., 2006; Duchi et al., 2010; Shalev-Shwartz, 2012). Here we review the most relevant works with a focus on the dynamic regret.

### 5.1 DYNAMIC REGRET OF ONLINE GRADIENT DESCENT

Dynamic regret was first introduced in (Zinkevich, 2003) for the analysis of online gradient descent, where an $O(\sqrt{T}C_T)$ bound on the dynamic regret was derived for convex functions. When the learner has knowledge of the path length beforehand, the dynamic regret can be upper bounded by $O(\sqrt{TC_T})$

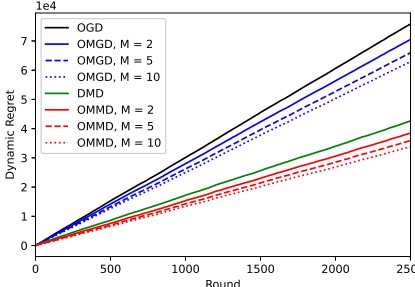 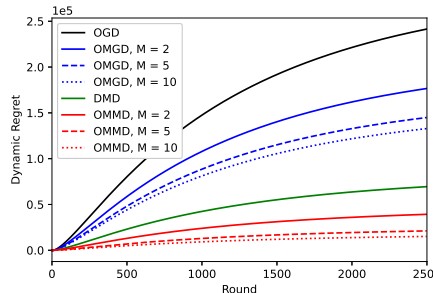

Figure 1: Dynamic regret comparison on CIFAR-10 dataset.

Figure 2: Dynamic regret of slowly drifting cost functions.

(Yang et al., 2016). For strongly convex cost functions, the upper bound on the dynamic regret can be reduced to $O(C_T)$ (Mokhtari et al., 2016). The above works make only a single query to the gradient of the cost functions in every round. By allowing the learner to make multiple gradient queries per online round, the regret bound can be improved to $O(\min(C_T, S_T))$ when the cost functions are smooth and strongly convex (Zhang et al., 2017). The analysis in all aforementioned studies requires the cost functions to be Lipschitz continuous. However, many commonly used cost functions do not satisfy this condition over an unbounded feasible set, e.g., the quadratic function, and even when the feasible set is bounded the Lipschitz factor can be excessively large, especially when the cost functions are strongly convex. Therefore, Lipschitz continuity of the cost functions is not assumed in our analysis. Instead, we move this condition from the cost functions to the Bregman divergence, which the learner can control and design. In addition, the above works rely on measuring distances using Euclidean norms, while the updates with Euclidean distance are challenging for some constraint sets, e.g., probability simplex (Duchi, 2018). It is known that gradient descent does not perform as well as mirror descent, especially when the input dimension is high (Nemirovsky & Yudin, 1983; Beck & Teboulle, 2003).

## 5.2 DYNAMIC REGRET OF ONLINE MIRROR DESCENT

The dynamic regret of online single-step mirror descent was studied in (Hall & Willett, 2015), where an upper bound of $O(\sqrt{T}(1 + C_T))$ was derived for convex cost functions. To take advantage of smoothness in cost functions, an adaptive algorithm based on optimistic mirror descent (Rakhlin & Sridharan, 2013) was proposed in (Jadbabaie et al., 2015), which contains two steps of mirror descent per online round. However, different from our work, in that variant the learner is allowed to make only a single query about the gradient. The algorithm further requires some prior prediction of the gradient in each round, which is used in the second mirror descent step. The dynamic regret bound was given in terms of a combination of the path length $C_T$, deviation between the predictions and the actual gradients $D_T$, and functional variation $F_T = \sum_{t=1}^{T} \max_{x \in \mathcal{X}} |f_t(x) - f_{t-1}(x)|$. [2] Unfortunately, to achieve this bound, the algorithm requires the design of a time-varying step size that depends on the optimal solution in the previous step, which prevents direct numerical comparison with OMMD. Therefore, in Section 4 we have experimented only with the method of (Hall & Willett, 2015).

All aforementioned works make only a single query to the gradient of the cost functions in every online round. In contrast, in this work, we allow the learner to make multiple gradient queries per round. The learner then uses this information to update its decision via multiple steps of mirror descent. In this way, we show the dynamic regret can be upper bounded linearly by the minimum among the path length, squared path length, and sum squared gradients. Furthermore, as opposed to the aforementioned works, our analysis does not require the cost functions to be Lipschitz continuous.

Finally, there is also recent work in the literature on distributed online mirror descent (Shahrampour & Jadbabaie, 2017). As expected, it is more challenging to achieve performance guarantee in distributed optimization. We focus on centralized online convex optimization in this work.

---

[2] We note that the regret bounds derived in (Jadbabaie et al., 2015) is under the same definition of (Zinkevich, 2003).

## 6 CONCLUSION

We have studied online convex optimization in dynamic settings. By applying the mirror descent step multiple times in each round, we show that the upper bound on the dynamic regret can be reduced significantly from $O(\sqrt{T}(1 + C_T))$ to $O(\min(C_T, S_T, G_T))$, when the cost functions are strongly convex and smooth. In contrast to prior studies (Hall & Willett, 2015; Zhang et al., 2017; Mokhtari et al., 2016), our analysis does not require the cost functions to be Lipschitz continuous. Numerical experiments with the CIFAR-10 dataset, and sequential quadratic programming, and additional examples show substantial improvement on the dynamic regret compared with existing alternatives.

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

## A ADDITIONAL DEFINITIONS

*Definition 3*: A function $f(\cdot)$ is $L$-smooth, if there exists a positive constant $L$ such that

$$f(y) \leq f(x) + \langle \nabla f(x), y - x \rangle + \frac{L}{2}\|y - x\|^2, \ \forall x, y \in \mathcal{X}.$$

*Definition 4*: A convex function $f(\cdot)$ is $\lambda$-strongly convex with respect to some norm $\|\cdot\|$, if there exists a positive constant $\lambda$ such that

$$f(y) + \langle \nabla f(y), x - y \rangle + \frac{\lambda}{2}\|x - y\|^2 \leq f(x), \ \forall x, y \in \mathcal{X}.$$

*Definition 5*: A function $f(\cdot)$ is Lipschitz continuous with factor $G$ if for all $x$ and $y$ in $\mathcal{X}$, the following holds:

$$|f(x) - f(y)| \leq G\|x - y\|, \ \forall x, y \in \mathcal{X}.$$

## B PROOF OF LEMMA 1

Consider single-step mirror descent update as follows:

$$x = \operatorname*{argmin}_{y \in \mathcal{X}} \left\{ f_t(x') + \langle \nabla f_t(x'), y - x' \rangle + \frac{1}{\alpha} D_r(y, x') \right\}. \tag{7}$$

Strong convexity of the above minimization objective implies

$$f_t(x') + \langle \nabla f_t(x'), x - x' \rangle + \frac{1}{\alpha} D_r(x, x') \leq \tag{8}$$

$$f_t(x') + \langle \nabla f_t(x'), y - x' \rangle + \frac{1}{\alpha} D_r(y, x') - \frac{1}{\alpha} D_r(y, x), \ \forall y \in \mathcal{X}.$$

Furthermore, from the smoothness condition, we have

$$f_t(x) \leq f_t(x') + \langle \nabla f_t(x'), x - x' \rangle + \frac{L}{2}\|x - x'\|^2. \tag{9}$$

Substituting equation 9 into equation 8, and setting $y = x_t^*$, we obtain

$$f_t(x) - \frac{L}{2}\|x - x'\|^2 + \frac{1}{\alpha} D_r(x, x') \leq \tag{10}$$

$$f_t(x') + \langle \nabla f_t(x'), x_t^* - x' \rangle + \frac{1}{\alpha} D_r(x_t^*, x') - \frac{1}{\alpha} D_r(x_t^*, x).$$

Since $\alpha \leq \frac{1}{L}$, and regularization function $r(\cdot)$ is 1-strongly convex, we have

$$\frac{1}{\alpha} D_r(x, x') \geq L D_r(x, x') \geq \frac{L}{2}\|x - x'\|^2. \tag{11}$$

Next, we exploit the strong convexity of the cost function, i.e.,

$$f_t(x') + \langle \nabla f_t(x'), x_t^* - x' \rangle \leq f_t(x_t^*) - \lambda D_r(x_t^*, x'). \tag{12}$$

Combining equation 10, equation 11, and equation 12, we obtain

$$f_t(x) \leq f_t(x_t^*) - \lambda D_r(x_t^*, x') + \frac{1}{\alpha} D_r(x_t^*, x') - \frac{1}{\alpha} D_r(x_t^*, x). \tag{13}$$

Next, we use the result of (Hazan & Kale, 2014), which states that for evey $\lambda$-strongly convex function $f_t(.)$, the following bound holds:

$$f_t(x) - f_t(x_t^*) \geq \lambda D_r(x_t^*, x), \tag{14}$$

where $x_t^* = \operatorname{argmin}_{x \in \mathcal{X}} f_t(x)$. Combining the above with equation 13, we obtain

$$D_r(x_t^*, x) \leq \beta D_r(x_t^*, x'), \tag{15}$$

where $\beta = 1 - \frac{2\lambda\alpha}{1 + \lambda\alpha}$. □

## C    PROOF OF THEOREM 2

### C.1    KEY LEMMAS

The following two lemmas pave the way for our regret analysis leading to Theorem 2. Lemma 7 presents an alternative form for the mirror descent update.

**Lemma 7** *Suppose there exists $z_{t+1}$ that satisfies $\nabla r(z_{t+1}) = \nabla r(x_t) - \alpha \nabla f_t(x_t)$, for some strongly convex function $r(\cdot)$, and step size $\alpha$. Then, the following updates are equivalent*

$$x_{t+1} = \operatorname*{argmin}_{x \in \mathcal{X}} D_r(x, z_{t+1}), \tag{16}$$

$$x_{t+1} = \operatorname*{argmin}_{x \in \mathcal{X}} \left\{ \langle \nabla f_t(x_t), x \rangle + \frac{1}{\alpha} D_r(x, x_t) \right\}. \tag{17}$$

*Proof.* We begin by expanding equation 16 as follows:

$$
\begin{aligned}
x_{t+1} &= \operatorname*{argmin}_{x \in \mathcal{X}} \{ r(x) - r(z_{t+1}) - \langle \nabla r(z_{t+1}), x - z_{t+1} \rangle \} \\
&= \operatorname*{argmin}_{x \in \mathcal{X}} \{ r(x) - \langle \nabla r(z_{t+1}), x \rangle \} \\
&= \operatorname*{argmin}_{x \in \mathcal{X}} \{ r(x) - \langle \nabla r(x_t) - \alpha \nabla f_t(x_t), x \rangle \} \\
&= \operatorname*{argmin}_{x \in \mathcal{X}} \{ \alpha \langle \nabla f_t(x_t), x \rangle + r(x) - r(x_t) - \langle \nabla r(x_t), x - x_t \rangle \} \\
&= \operatorname*{argmin}_{x \in \mathcal{X}} \{ \langle \nabla f_t(x_t), x \rangle + \frac{1}{\alpha} D_r(x, x_t) \}.
\end{aligned}
\tag{18}
$$

Thus, the update in equation 16 is equivalent to equation 17.    □

**Lemma 8** *Under the same convexity and smoothness condition stated in Theorem 2, let $x_t$ be the sequence of decisions generated by OMMD. Then, the following bound holds:*

$$\|x_{t+1} - x_t^*\| \le \sqrt{L_r \beta^M} \|x_t - x_t^*\|, \tag{19}$$

*where $L_r$ is the smoothness factor of the regularization function $r(\cdot)$, and $\beta$ is the shrinking factor obtained in Lemma 1.*

*Proof.* Using the result of Lemma 1, OMMD with $M$ mirror descent steps guarantees

$$D_r(x_t^*, x_{t+1}) \le \beta^M D_r(x_t^*, x_t). \tag{20}$$

Since the regularization function $r(\cdot)$ is 1-strongly convex, we have

$$\frac{\|x_t^* - x_{t+1}\|^2}{2} \le r(x_t^*) - r(x_{t+1}) - \langle \nabla r(x_{t+1}), x_t^* - x_{t+1} \rangle. \tag{21}$$

Next, we exploit the smoothness condition of the regularization function $r(\cdot)$, i.e.,

$$r(x_t^*) - r(x_t) - \langle \nabla r(x_t), x_t^* - x_t \rangle \le \frac{L_r}{2} \|x_t^* - x_t\|^2. \tag{22}$$

By combining the above with equation 20, and equation 21, and using the definition of Bregman divergence, we obtain

$$\|x_{t+1} - x_t^*\|^2 \le L_r \beta^M \|x_t - x_t^*\|^2. \tag{23}$$

Taking the square root on both sides of equation 23 completes the proof.    □

## C.2 PROOF OF THE THEOREM

Now, we are ready to present the proof of Theorem 2. In this proof, we will use the following properties of Bregman divergence.

(a) By direct substitution, the following equality holds for any $x, y, z \in \mathcal{X}$,

$$\langle \nabla r(z) - \nabla r(y), x - y \rangle = D_r(x, y) - D_r(x, z) + D_r(y, z). \tag{24}$$

(b) If $x = \operatorname{argmin}_{x' \in \mathcal{X}} D_r(x', z)$, i.e., $x$ is the Bregman projection of $z$ into the set $\mathcal{X}$, then for any arbitrary point $y \in \mathcal{X}$, we have

$$D_r(y, z) \geq D_r(y, x) + D_r(x, z). \tag{25}$$

To bound the dynamic regret, we begin by using the strong convexity of the cost function $f_t(\cdot)$, i.e.,

$$
\begin{aligned}
f_t(x_t) - f_t(x_t^*) &\leq \langle \nabla f_t(x_t), x_t - x_t^* \rangle - \lambda D_r(x_t^*, x_t) \\
&\leq \frac{1}{\alpha} \langle \nabla r(x_t) - \nabla r(z_{t+1}), x_t - x_t^* \rangle - \lambda D_r(x_t^*, x_t) \\
&\leq \frac{1}{\alpha} \Big( D_r(x_t^*, x_t) - D_r(x_t^*, z_{t+1}) + D_r(x_t, z_{t+1}) \Big) - \lambda D_r(x_t^*, x_t) \\
&\leq \frac{1}{\alpha} \Big( D_r(x_t^*, x_t) - D_r(x_t^*, x_{t+1}) - D_r(x_{t+1}, z_{t+1}) + D_r(x_t, z_{t+1}) \Big) - \lambda D_r(x_t^*, x_t) \\
&\leq \Big( \frac{1}{\alpha} - \lambda \Big) D_r(x_t^*, x_t) + \frac{1}{\alpha} \Big( D_r(x_t, z_{t+1}) - D_r(x_{t+1}, z_{t+1}) \Big) \\
&\leq \Big( \frac{1}{\alpha} - \lambda \Big) \Big( \frac{f_t(x_t) - f_t(x_t^*)}{\lambda} \Big) + \frac{1}{\alpha} \Big( D_r(x_t, z_{t+1}) - D_r(x_{t+1}, z_{t+1}) \Big), \tag{26}
\end{aligned}
$$

where in the second line we have used the alternative mirror descent update stated in Lemma 7, i.e., $\nabla f_t(x_t) = (1/\alpha)(\nabla r(x_t) - \nabla r(z_{t+1}))$. To obtain the third line, we have utilized the Bregman divergence property in equation 24. We have used the Bregman projection property in equation 25 in the fourth line. By omitting some negative terms, and using equation 14, we obtain the right-hand side of equation 26.

Thus, if $\alpha > \frac{1}{2\lambda}$, we have

$$
\begin{aligned}
f_t(x_t) - f_t(x_t^*) &\leq \frac{\lambda}{2\alpha\lambda - 1} \Big( D_r(x_t, z_{t+1}) - D_r(x_{t+1}, z_{t+1}) \Big) \\
&\overset{(a)}{\leq} \frac{\lambda K}{2\alpha\lambda - 1} \|x_{t+1} - x_t\| \\
&\leq \frac{\lambda K}{2\alpha\lambda - 1} \Big( \|x_{t+1} - x_t^*\| + \|x_t - x_t^*\| \Big) \\
&\overset{(b)}{\leq} \frac{\lambda K}{2\alpha\lambda - 1} (1 + \sqrt{L_r \beta M}) \|x_t - x_t^*\|, \tag{27}
\end{aligned}
$$

where we have used the Lipschitz continuity of Bregman divergence to obtain inequality (a), and we have applied Lemma 8 to obtain inequality (b). Summing equation 27 over time, we have

$$\text{Reg}_T^d = \sum_{t=1}^T f_t(x_t) - f_t(x_t^*) \leq \frac{\lambda K}{2\alpha\lambda - 1} (1 + \sqrt{L_r \beta M}) \sum_{t=1}^T \|x_t - x_t^*\|. \tag{28}$$

Now, we proceed to bound $\sum_{t=1}^T \|x_t - x_t^*\|$ as follows:

$$
\begin{aligned}
\sum_{t=1}^T \|x_t - x_t^*\| &= \|x_1 - x_1^*\| + \sum_{t=2}^T \|x_t - x_t^*\| \\
&\leq \|x_1 - x_1^*\| + \sum_{t=2}^T \|x_t - x_{t-1}^*\| + \|x_{t-1}^* - x_t^*\| \\
&\overset{(a)}{\leq} \|x_1 - x_1^*\| + \sum_{t=2}^T \sqrt{L_r \beta M} \|x_{t-1} - x_{t-1}^*\| + \sum_{t=2}^T \|x_t^* - x_{t-1}^*\|, \tag{29}
\end{aligned}
$$

where we used the result of Lemma 8 to obtain inequality (a). If $M \geq \lceil \left( \frac{1}{2} + \frac{1}{2\alpha\lambda} \right) \log L_r \rceil$, we have

$$\beta^M = \left( 1 - \frac{2\alpha\lambda}{1+\alpha\lambda} \right)^M \leq \exp\left( \frac{-2M\alpha\lambda}{1+\alpha\lambda} \right) < \frac{1}{L_r}, \tag{30}$$

which implies $L_r \beta^M < 1$. Therefore, by combining equation 29 and equation 30, we have

$$\sum_{t=1}^{T} \|x_t - x_t^*\| \leq \frac{\|x_1 - x_1^*\|}{1 - \sqrt{L_r \beta^M}} + \frac{\sum_{t=2}^{T} \|x_t^* - x_{t-1}^*\|}{1 - \sqrt{L_r \beta^M}}. \tag{31}$$

Finally, substituting equation 31 into equation 28 completes the proof. $\qquad\square$

## D PROOF OF THEOREM 3

In order to bound the dynamic regret, we begin by the smoothness condition of the cost function $f_t(.)$, i.e.,

$$f_t(x_t) - f_t(x_t^*) \leq \langle \nabla f_t(x_t^*), x_t - x_t^* \rangle + \frac{L}{2}\|x_t - x_t^*\|^2$$

$$\leq \|\nabla f_t(x_t^*)\|_* \|x_t - x_t^*\| + \frac{L}{2}\|x_t - x_t^*\|^2. \tag{32}$$

Next, we use the fact

$$\|\nabla f_t(x_t^*)\|_* \|x_t - x_t^*\| \leq \frac{\|\nabla f_t(x_t^*)\|_*^2}{2\theta} + \frac{\theta\|x_t - x_t^*\|^2}{2}, \tag{33}$$

for any arbitrary positive constant $\theta > 0$. Thus, we have

$$f_t(x_t) - f_t(x_t^*) \leq \frac{\|\nabla f_t(x_t^*)\|_*^2}{2\theta} + \frac{(L+\theta)\|x_t - x_t^*\|^2}{2}. \tag{34}$$

Summing equation 34 over time, we obtain

$$\text{Reg}_T^d = \sum_{t=1}^{T} f_t(x_t) - f_t(x_t^*) \leq \sum_{t=1}^{T} \frac{\|\nabla f_t(x_t^*)\|_*^2}{2\theta} + \frac{L+\theta}{2} \sum_{t=1}^{T} \|x_t - x_t^*\|^2. \tag{35}$$

Now, we proceed by bounding $\sum_{t=1}^{T} \|x_t - x_t^*\|^2$ as follows:

$$\sum_{t=1}^{T} \|x_t - x_t^*\|^2 = \|x_1 - x_1^*\|^2 + \sum_{t=2}^{T} \|x_t - x_{t-1}^* + x_{t-1}^* - x_t^*\|^2$$

$$\leq \|x_1 - x_1^*\|^2 + \sum_{t=2}^{T} \left( 2\|x_t - x_{t-1}^*\|^2 + 2\|x_{t-1}^* - x_t^*\|^2 \right)$$

$$\leq \|x_1 - x_1^*\|^2 + 2\beta^M L_r \sum_{t=1}^{T} \|x_t - x_{t-1}^*\|^2 + 2\sum_{t=2}^{T} \|x_{t-1}^* - x_t^*\|^2. \tag{36}$$

We note that if $M \geq \lceil \left( \frac{1}{2} + \frac{1}{2\alpha\lambda} \right) \log 2L_r \rceil$, then $2\beta^M L_r < 1$. Therefore, from equation 36 we can obtain

$$\sum_{t=1}^{T} \|x_t - x_t^*\|^2 \leq \frac{\|x_1 - x_1^*\|^2}{1 - 2\beta^M L_r} + \frac{2}{1 - 2\beta^M L_r} \sum_{t=2}^{T} \|x_t^* - x_{t-1}^*\|^2. \tag{37}$$

Substituting equation 37 into equation 35 completes the proof. $\qquad\square$

## E  PROOF OF THEOREM 5

The proof of Theorem 5 initially follows the first half of the proof of Theorem 2, which is repeated here for completeness.

To analyze the dynamic regret, we first use the strong convexity of the cost function $f_t(\cdot)$, i.e.,

$$
\begin{aligned}
f_t(x_t) - f_t(x_t^*) &\leq \langle \nabla f_t(x_t), x_t - x_t^* \rangle - \lambda D_r(x_t^*, x_t) \\
&\leq \frac{1}{\alpha} \langle \nabla r(x_t) - \nabla r(z_{t+1}), x_t - x_t^* \rangle - \lambda D_r(x_t^*, x_t) \\
&\leq \frac{1}{\alpha} \Big( D_r(x_t^*, x_t) - D_r(x_t^*, z_{t+1}) + D_r(x_t, z_{t+1}) \Big) - \lambda D_r(x_t^*, x_t) \\
&\leq \frac{1}{\alpha} \Big( D_r(x_t^*, x_t) - D_r(x_t^*, x_{t+1}) - D_r(x_{t+1}, z_{t+1}) + D_r(x_t, z_{t+1}) \Big) - \lambda D_r(x_t^*, x_t) \\
&\leq \Big( \frac{1}{\alpha} - \lambda \Big) D_r(x_t^*, x_t) + \frac{1}{\alpha} \Big( D_r(x_t, z_{t+1}) - D_r(x_{t+1}, z_{t+1}) \Big) \\
&\leq \Big( \frac{1}{\alpha} - \lambda \Big) \Big( \frac{f_t(x_t) - f_t(x_t^*)}{\lambda} \Big) + \frac{1}{\alpha} \Big( D_r(x_t, z_{t+1}) - D_r(x_{t+1}, z_{t+1}) \Big), \quad (38)
\end{aligned}
$$

where in the second line we have used the alternative mirror descent update stated in Lemma 7, i.e., $\nabla f_t(x_t) = (1/\alpha)(\nabla r(x_t) - \nabla r(z_{t+1}))$. To obtain the third line, we have utilized the Bregman divergence property in equation 24. We have used the Bregman projection property in equation 25 in the fourth line. By omitting some negative terms, and using equation 14, we obtain the right-hand side of equation 38.

Therefore, if $\alpha > \frac{1}{2\lambda}$, we have

$$
f_t(x_t) - f_t(x_t^*) \leq \frac{\lambda}{2\alpha\lambda - 1} \left( D_r(x_t, z_{t+1}) - D_r(x_{t+1}, z_{t+1}) \right). \quad (39)
$$

Now we continue to bound $D_r(x_t, z_{t+1})$. By the definition of Bregman divergence, we have

$$
\begin{aligned}
D_r(x_t, z_{t+1}) + D_r(z_{t+1}, x_t) &= \langle \nabla r(x_t) - \nabla r(z_{t+1}), x_t - z_{t+1} \rangle \\
&= \langle \alpha \nabla f_t(x_t), x_t - z_{t+1} \rangle \\
&\leq \|\alpha \nabla f_t(x_t)\|_* \|x_t - z_{t+1}\| \\
&\leq \frac{\alpha^2}{2} \|\nabla f_t(x_t)\|_*^2 + \frac{\|x_t - z_{t+1}\|^2}{2}. \quad (40)
\end{aligned}
$$

The strong convexity of the regularization function implies

$$
\frac{\|x_t - z_{t+1}\|^2}{2} \leq r(z_{t+1}) - r(x_t) - \langle \nabla r(x_t), z_{t+1} - x_t \rangle = D_r(z_{t+1}, x_t). \quad (41)
$$

Combining the above with equation 40, we obtain

$$
D_r(x_t, z_{t+1}) \leq \frac{\alpha^2}{2} \|\nabla f_t(x_t)\|_*^2. \quad (42)
$$

By substituting equation 42 into equation 40, and summing over time, we have

$$
\text{Reg}_T^d = \sum_{t=1}^{T} f_t(x_t) - f_t(x_t^*) \leq \frac{\alpha^2 \lambda}{4\alpha\lambda - 2} \|\nabla f_t(x_t)\|_*^2. \quad (43)
$$

$\square$

## F  CLOSED-FORM UPDATE FOR MIRROR DESCENT

In this section, we derive the close-form mirror descent update in equation 6.

Let $r(y) = \sum_{j=1}^{d} y_j \log(y_j)$ be the negative entropy. Then, we have

$$D_r(y, y_t^i) = \sum_{j=1}^{d} \left[ y_j \log(y_j) - y_{t,j}^i \log(y_{t,j}^i) - (\log(y_{t,j}^i) + 1)(y_j - y_{t,j}^i) \right]$$

$$= \sum_{j=1}^{d} y_j \log(\frac{y_j}{y_{t,j}^i}) + \langle 1, y - y_t^i \rangle = D_{KL}(y, y_t^i), \tag{44}$$

where $y_{t,j}^i$ denotes the $j$-th component of the decision vector $y_t^i$, and $D_{KL}(y, y_t^i)$ represents the KL divergence between $y$ and $y_t^i$.

Now consider the update in equation 5, which can be written as follows:

$$\text{minimize}_{y \in \mathcal{X}} \quad \langle \nabla f_t(y_t^i), y \rangle + \frac{1}{\alpha} \sum_{j=1}^{d} y_j \log(\frac{y_j}{y_{t,j}^i}) \tag{45}$$

$$\text{subject to} \quad \langle 1, y \rangle = 1, y \geq 0.$$

The Lagrangian of the above problem is given by

$$L(y, \lambda, \gamma) = \langle \nabla f_t(y_t^i), y \rangle + \sum_{j=1}^{d} \left[ \frac{1}{\alpha} y_j \log(\frac{y_j}{y_{t,j}^i}) + \lambda y_j - \gamma_j y_j \right] - \lambda, \tag{46}$$

where $\lambda \in \mathbb{R}$ and $\gamma \in \mathbb{R}_+^d$ are Lagrange multipliers corresponding to the constraints. Next, we take derivative with respect to $y$ to obtain

$$\frac{\partial}{\partial y_j} L(y, \lambda, \gamma) = \nabla f_t(y_t^i)_j + \frac{1}{\alpha} \log(y_j) + \frac{1}{\alpha} - \frac{1}{\alpha} \log(y_{t,j}^i) + \lambda - \gamma_j. \tag{47}$$

Setting the above to zero results in the following closed-form update:

$$y_{t,j}^{i+1} = \frac{y_{t,j}^i \exp(-\alpha \nabla f_t(y_{t,j}^i))}{\sum_{j=1}^{d} y_{t,j}^i \exp(-\alpha \nabla f_t(y_{t,j}^i))}. \tag{48}$$

$\square$

# G  ADDITIONAL EXPERIMENTS

In this section, we present additional experiments to study the performance of OMMD. In the first experiment, we use the MNIST dataset. In the second experiment, we consider a switching problem where the cost function switches between two quadratic functions after a specific number of rounds.

First, we consider the well-known MNIST digits dataset, where every data sample $\omega$ is an image of size $28 \times 28$ pixel that can be represented by a 784-dimensional vector, i.e., $d = 784$. Each sample corresponds to one of the digits in $\{0, 1, \ldots, 9\}$, and thus, there are $c = 10$ different classes. The goal of the learner is to classify streaming digit images in an online fashion.

We consider a robust regression problem, where the cost function for the batch of data samples at time $t$ is given by

$$f(x, (\omega_t, z_t)) = \|\omega_t^T x - z_t\|_1^2,$$

where $x$ is the optimization variable, belonging to the constraint set is $\mathcal{X} = \{x : x \in \mathbb{R}_+^n, \| x \|_1 = 1\}$. We use the negative entropy regularization function, i.e., $r(x) = \sum_{i=1}^{d} x_i \log(x_i)$, which is strongly convex with respect to the $l1-$norm. We set the step size $\alpha = 0.1$ and use a batch size of 20 data examples per round.

From Fig. 3, we again observe that OMMD consistently outperforms the other alternatives. In particular, compared with DMD, applying $M = 10$ steps of mirror descent can reduce the dynamic regret up to $20\%$. We also see that the dynamic regret grows linearly with the number of rounds,

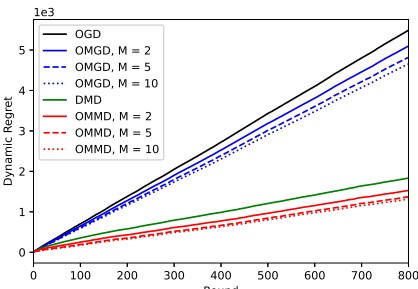 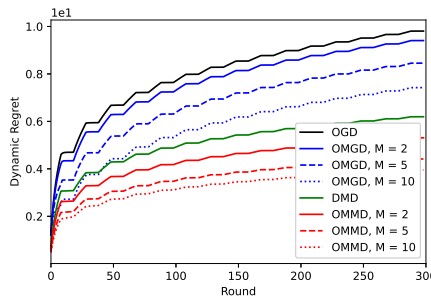

Figure 3: Dynamic regret comparison on MNIST dataset.

Figure 4: Dynamic regret comparison for switching cost.

which is a natural consequence of steady fluctuation in the sequence of dynamic minimizers $x_t^*$ as explained before.

Next, we consider the case where the cost function switches between two functions. Both functions are in the quadratic form $f_t(x) = \|A_t x - b_t\|_2^2$, where $A_t \in \mathbb{R}^{d \times d}$, and $b_t \in \mathbb{R}^d$. In particular, we assume that the parameter $A_t$ is chosen among

$$A_t^{(1)} = \text{diag}(\underbrace{\frac{1}{t^{p_1}}, \frac{1}{t^{p_1}}, \ldots, \frac{1}{t^{p_1}}}_{d_1}, \underbrace{0, 0, \ldots, 0}_{d_2}), \text{ and } A_t^{(2)} = \text{diag}(\underbrace{0, 0, \ldots, 0}_{d_1}, \underbrace{\frac{1}{t^{p_1}}, \frac{1}{t^{p_1}}, \ldots, \frac{1}{t^{p_1}}}_{d_2}),$$

such that $d_1 + d_2 = d$, and $b_t = [\frac{1}{t^{p_2}}, \ldots, \frac{1}{t^{p_2}}]'$. Therefore, at each round the cost function is either $f_t^{(1)}(x) = \|A_t^{(1)} x - b_t\|_2^2$ or $f_t^{(2)}(x) = \|A_t^{(2)} x - b_t\|_2^2$. We assume that the cost function switches between $f_t^{(1)}(\cdot)$ and $f_t^{(2)}(\cdot)$ every $\tau$ rounds. In our experiment, we set $d_1 = 10$, $d = 1000$, $p_1 = 0.9$, and $p_2 = 0.1$. We further set the switching period $\tau = 10$, and parameter $\alpha = 0.02$. The dynamic regret roughly reflects the accumulated mismatch error over time.

In Fig. 4, we compare the performance of OMMD with that of other alternatives in terms of the dynamic regret. OMMD with $M = 10$ nearly halves the dynamic regret of DMD after 300 rounds. Furthermore, the benefit of applying multiple steps of mirror descent can be significant even for smaller values of $M$.

