# OpenReview forum: "On the Dynamic Regret of Online Multiple Mirror Descent"
_ICLR.cc/2021/Conference — Reject_

### Official Review · AnonReviewer4 · 2020-10-26
**The significance of the contribution is unclear**

**Rating:** 4
**Confidence:** 4

**Review:**

This work derives a new upper bound on the dynamic regret for online convex optimization in the restricted setting where the comparison sequence is made up of the minimizers x^_1,...,x*_T of the loss sequence. There are three main parameters that control regret in this case: the path length C_T = ||x*_2-x*_1||+...+||x*_T-x*_{T-1}||. The squared path length C_T = ||x*_2-x*_1||^2+...+||x*_T-x*_{T-1}||^2. And the sum G_T of squared loss gradient norms evaluated at x*_1,...,x*_T.

Previous work by Zhang showed that when the loss functions are simultaneously strongly convex and strongly smooth w.r.t. the Euclidean norm, then Online Gradient Descent with multiple updates per step (OMGD) achieves regret of order min{C_T,S_T} whenever G_T < T G^2 for some G > 0 (i.e., the gradient norm of the loss is uniformly bounded). Note that this algorithm invokes the first-order oracle a constant number of times per step.

This work extends Zhang's results in two main directions. First, OMGD is replaced by OMMD (which is Online Multiple Mirror Descent). Accordingly, strong convexity and strong smoothness are now computed w.r.t. the Bregman divergence induced by the mirror map. Second, the condition G_T < T G^2 is dropped. The resulting bound is of order min{C_T,S_T,G_T}. Experiments with quadratic losses compare OMMD against OMGD and Dynamic Mirror Descent (DMD) by Hall and Willet. Both these algorithms have worse upper bounds than OMMD.

I have some concerns about the significance of the results. I wonder what are interesting examples of loss functions that are simultaneously strongly convex and strongly smooth with respect to a Lipschitz continuous Bregman divergence induced by a regularization function which ---in turn--- is simultaneously strongly convex and strongly smooth with respect to some norm? We know that quadratic losses and regularized logistic losses satisfy these assumptions w.r.t the squared Euclidean mirror map (as shown by Zhang). I suspect that the generalization proposed in this work does not lead to include additional interesting losses.

Since OMMD improves on the OMGD regret bound, it would be interesting to know whether OMMD reduces to OMGD when the regularizer is Euclidean.

The experiments use a decision set (the simplex) where OMD with entropic regularization is known to work better than OGD. This advantage appears to be simply transferred to the dynamic setting. Indeed, OMMD performs only marginally better than DMD, which also uses the entropic regularization.

It is not even clear whether the theory applies to the setting used in the experiments. The quadratic losses used in the experiments are strongly convex and smooth only with respect to Euclidean regularization and not with respect to entropic regularization (the regularization used by OMMD in the experiments).

The assumptions require Lipschitz continuity of the Bregman divergence. Does this condition apply to KL divergence? What is K in this case?

In the ridge regression experiment, z_i is defined as a scalar but treated as a vector in the definition of the loss.

---

> ### Author Response · Authors · 2020-11-17
> **Response to Reviewer 4 (1/2)**
>
> We thank the reviewer for his/her helpful comments.
>
> First, we would like to clarify that (Zhang et al., 2017) derived dynamic regret bounds only based on the path length $C_T$ and squared path length $S_T$. The definition of $G_T$ was $\text{not}$ given in (Zhang et al., 2017). Their analysis requires the Lipschitz condition ($||\nabla f(x)||\leq G$), which given our definition of $G_T$, does imply $G_T \leq T G^2$. In this work, we have removed the Lipschitz continuity assumption on the cost functions. Therefore, our new regret bounds based on $C_T$ and $S_T$ (which require  different derivation from those in (Zhang et. al, 2017)) are applicable to a broader range of problems. Overall,  the resulting upper bound of $O(\min({C_T,S_T,G_T}))$ is stronger than the regret bounds of $O(\sqrt{T}(1+C_T) )$ and $O(\min{(C_T,S_T)})$ associated with DMD and OMGD, respectively.
>
> Q1 - I have some concerns about the significance of the results. I wonder what are interesting examples of loss functions that are simultaneously strongly convex and strongly smooth with respect to a Lipschitz continuous Bregman divergence induced by a regularization function which ---in turn--- is simultaneously strongly convex and strongly smooth with respect to some norm? We know that quadratic losses and regularized logistic losses satisfy these assumptions w.r.t the squared Euclidean mirror map (as shown by Zhang et al.). I suspect that the generalization proposed in this work does not lead to include additional interesting losses.
>
> A1 - The classical definitions of Lipschitz continuity, smoothness, and strongly convexity limits the applicability of many prior studies on the performance of optimization methods. This has motivated the generalization of these definitions in recent studies (Bauschke et al., 2017; Lu et al., 2018; Antonakopoulos et al., 2020; Zhou et al., 2020). In particular, in the literature of optimization methods, there is wide adoption of the Bregman divergence in the definition of strongly convex functions, instead of the Euclidean norm used in gradient descent (Shalev-Schwartz et al., 2007; J. Duchi et al., 2010; Y. Lei et al., 2018).  The Lipschitz continuity of the Bregman divergence is also commonly assumed in studies of online mirror descent (Jadbabaie et al, 2015; Shahrampour et al., 2017).
> There are many mathematical examples illustrating that while a class of cost functions are not strongly convex (resp. smooth) with respect to the squared Euclidean norm, they still can be considered strongly convex (resp. smooth) under a more general definition (Bauschke et al., 2017). For example, there are practical problems in imaging science such as the Poisson inverse problem, where the corresponding cost function does not have the aforementioned classically defined properties. This problem deals with recovering a high-dimensional image represented by a vector with non-negative entries, where the KL divergence is a popular regularizer to measure the proximity of such vectors. We note that, by Pinsker's inequality, the KL divergence is strongly convex with respect to the $l$-1 norm. The KL divergence is also Lipschitz continuous as explained in our response to Q5. The class of problems that are not covered by the analysis with classical assumptions is sufficiently broad to illustrate the importance of our analysis in this work. Our purpose here is to provide a stronger theoretical guarantee for a larger class of realistic problems.
>
> Q2 - Since OMMD improves on the OMGD regret bound, it would be interesting to know whether OMMD reduces to OMGD when the regularizer is Euclidean.
>
> A2 - Mirror descent generalizes the gradient descent method by using the Bregman divergence with respect to some regularization function. Therefore, mirror descent contains gradient descent as a special case. In particular, choosing $r(x) = \frac{1}{2}||x||_2^2$ reduces the mirror descent update to the gradient descent update. Thus, any method that is based on mirror descent can be reduced to gradient descent with the special choice of the squared Euclidean norm as the regularization function.
>
> Q3 - The experiments use a decision set (the simplex) where OMD with entropic regularization is known to work better than OGD. This advantage appears to be simply transferred to the dynamic setting. Indeed, OMMD performs only marginally better than DMD, which also uses the entropic regularization.
>
> A3 - We note that OMMD $\text{substantially outperforms}$ DMD in the four experiments presented the paper and the supplementary material. In particular,  applying $M=10$ steps of OMMD can reduce the dynamic regret, in comparison with DMD, up to 30\%, 80\%, 20\%, and 50\% as shown in Figures 1, 2, 3, and 4, respectively. We respectfully contend that such improvement is well beyond marginal.

---

> > ### Author Response · Authors · 2020-11-17
> > **Response to Reviewer 4 (2/2)**
> >
> > Q4 - It is not even clear whether the theory applies to the setting used in the experiments. The quadratic losses used in the experiments are strongly convex and smooth only with respect to Euclidean regularization and not with respect to entropic regularization (the regularization used by OMMD in the experiments).
> >
> > A4 - The entropic regularizer is a common setup for the mirror descent method, e.g., see Section 4.2 of (J. Duchi, 2016). It is widely used since it leads to a closed-form update, which facilitates fast computation.
> > Therefore, our contribution in this work is two-fold in the following sense. First, under conditions that are more general than all prior studies, we confirm the theoretical advantage of OMMD over existing alternatives by deriving its dynamic regret bound based on $C_T$, $S_T$, and $G_T$. Second, through experiments, we show that OMMD substantially outperforms existing alternatives also in some popular settings that may not satisfy the conditions required in theoretical analysis.
> >
> > Q5 - The assumptions require Lipschitz continuity of the Bregman divergence. Does this condition apply to KL divergence? What is K in this case?
> >
> > A5 - The Lipschitz continuity of the Bregman divergence is commonly assumed in the analysis of OMD (Jadbabaie et.al., 2015, Shahrampour et al., 2017). For the KL divergence on the set $\mathcal{X} = {x | \sum_{i=1}^{d} x_i = 1; x_i \geq \frac{1}{D} }$, the constant $K$ is of $O(\log D)$.
> >
> > Q6 -  In the ridge regression experiment, $z_i$ is defined as a scalar but treated as a vector in the definition of the loss.
> >
> > A6 - In the ridge regression experiment, at each round a batch size of 20 is taken from the dataset. The parameter $z_i$ is a vector corresponding to the labels of the data examples in the batch. In the revision, we have added remarks to clarify this point.
> >
> > References cited above:
> >
> > H. H. Bauschke, J. Bolte, and M. Teboulle. A descent lemma beyond lipschitz gradient continuity: first-order methods revisited and applications. Mathematics of Operations Research, 42(2):330–306 348, 2017
> >
> > H. Lu, R. M. Freund, and Y. Nesterov. Relatively smooth convex optimization by first-order methods, and applications. SIAM Journal on Optimization, 28(1):333–354, 2018
> >
> > K. Antonakopoulos, E. V. Belmega, and P. Mertikopoulos. Online and stochastic optimization beyond lipschitz continuity: A riemannian approach. International Conference on Learning Representations, 2020
> >
> > S. Shalev-Schwartz. Logarithmic Regret Algorithms for Strongly Convex Repeated Games. Technical Report, The Hebrew University, 2007
> >
> > J. Duchi et. al. Composite objective mirror descent. In Proceedings of the Conference on Learning Theory, 2010
> >
> > Y. Lei et al. Stochastic composite mirror descent: optimal bounds with high probabilities. Advances in Neural Information Processing Systems. 2018
> >
> > A. Jadbabaie, et al. Online optimization:Competing with dynamic comparators. In Proceedings of the International Conference on Artificial Intelligence and Statistics, 2015
> >
> > S. Shahrampour et. al. Distributed online optimization in dynamic environments using mirror descent. IEEE Transactions on Automatic Control, 63(3):714–725, 2017
> >
> > J. Duchi. Introductory Lectures on Stochastic Convex Optimization. Park City Mathematics Institute, Graduate Summer School Lectures, July 2016

---

### Official Review · AnonReviewer2 · 2020-10-29
**Natural generalization of OMGD path-length regret; unclear justification of quantity G_T**

**Rating:** 5
**Confidence:** 4

**Review:**

Summary:
This paper studies the dynamic regret of online multiple mirror descent, which is online mirror descent with M repeated steps on each of T sequential loss functions. The authors show three bounds for the dynamic regret of OMMD, which generalizes OMGD [Zhang et al. '17]: C_T (the path length of the minimizer sequence), S_T (the sum of squared segment lengths), and G_T (the squared dual gradient norm of the points played).

Pros:
- This work fills a gap in the literature left by [Zhang et al. '17], which considers the analogous gradient descent algorithm, and derives the min(C_T, S_T) bound that this paper generalizes. It is natural to ask whether to corresponding mirror descent algorithm enjoys the benefits of mirror descent in classical online learning, in which the choice of regularizer allows for a statistical rate to adapt to the geometry of the decision set.
- The paper is well-written, and the analyses are crisp. There is a great deal of effort placed in including intuitive discussions along with the key results.

Cons:
- I'm not quite convinced by the motivation for G_T. Multiple points below in the detailed comments, where clarifications would be appreciated.
- Thus, overall, given my current understanding, I don't think this paper fully answers the research question in a way that is a sufficiently large delta compared to [Zhang et al. '17].

Detailed comments:
- Unlike C_T and S_T, G_T depends on the learner's decisions. So, if a learner makes very bad decisions and thus experiences large gradients (this is necessarily true by strong convexity), the G_T term becomes vacuous, and the regret bound is dominated by min(C_T, S_T).
- The discussion at the top of page 3 ("G_T can be smaller than both C_T and S_T, especially when the cost functions fluctuate drastically over time") is confusing: if cost functions fluctuate drastically over time, shouldn't the gradients experienced always be large? Seems like the opposite statement is the usual motivation for path length-based regret: "C_T and S_T can be smaller than G_T, if the functions fluctuate a lot but it's not worst-case because the minimizers are stable". A clarification, example, or mathematical statement would be helpful.
- Remark 4 is also confusing: if f needs to be strongly convex with a constant lambda, then its gradient dual norm can't shrink uniformly. Could the authors clarify?
- The fact that the Lipschitzness of the loss functions isn't assumed is also not necessarily a feature. At points where f_t is highly non-Lipschitz, the gradient is large, and thus G_T (the upper bound claimed to be the novelty) is large. While it's true that the regularizer r can be chosen more freely than the losses, a poor choice leads to large L_r, so it's not clear what claim that discussion (at the end of 2.1) is making.

*** post-response ***

Thanks for the clarifications and revisions in the manuscript.

- A2: Lemma 1 (and its M-step repetition) show that the outputs of mirror descent x_{t+1} are guaranteed to be close to the minimizers of f_t. But the loss of x_{t+1} is measured on f_{t+1}; this is what I meant by "bad decisions". Thus I'm confused about the statement "the learner is not expected to make bad decisions"; Lemma 1 does not imply this. It only implies this when the consecutive minimizers are close (which is captured by C_T and S_T being small).
- A3: This example makes sense (though in this example, the function is L-Lipschitz, not L-smooth). I also noticed that the example in Remark 5 isn't both smooth and strongly convex in any norm.
- A4: Understood; thanks.
- A5: I now understand (from the addendum in the paper as well) that this "user control of L_r" is basically referring to the standard selling point of OMD that the user can specify a regularizer that adapts to the geometry of the decision set and loss functions; I buy this point. This is separate from the ability to handle Lipschitz constants being large (which gradient descent can also handle, by having a smaller learning rate).

Additionally, I'm confused about the response to R2's "simple strategy". The convergence rates of GD and MD are stated in incompatible ways. The right comparison would be to pick M so that the RHS of Lemma 8 in the appendix to be smaller than \eps. I agree that this work strictly generalizes [Zhang et al. '17], but I don't agree that this work bypasses computational difficulties of approximately minimizing a smooth & strongly convex f_t; as the authors point out themselves in a different reply, when r is quadratic, OMMD reduces to running OMGD.

In my opinion, this algorithm and analysis are potentially worthy of publication at a top venue, but the manuscript evidently needs an overhaul in its justification of the setting and G_T. Thus my overall score is unchanged.

---

> ### Author Response · Authors · 2020-11-17
> **Response to Reviewer 2 (1/2)**
>
> Q1 - I'm not quite convinced by the motivation for $G_T$. Multiple points below in the detailed comments, where clarifications would be appreciated.
>
> A1 - We thank the reviewer for his/her helpful comments. We provide our response on this point under the detailed comments below.
>
> Q2 - Unlike $C_T$ and $S_T$, $G_T$ depends on the learner's decisions. So, if a learner makes very bad decisions and thus experiences large gradients (this is necessarily true by strong convexity), the $G_T$ term becomes vacuous, and the regret bound is dominated by $\min(C_T, S_T)$.
>
> A2 - Lemma 1 states that the learner's decision are close to the sequence of minimizers under the OMMD approach. Thus, the learner is not expected to make very bad decisions. We agree that sometimes $\min(C_T, S_T)$ can dominate $G_T$. , but we believe having an additional upper bound based on $G_T$ can be useful.
>
> Furthermore, we emphasize here that our contribution in this work includes all three upper bounds based on $C_T$, $S_T$, and $G_T$, and Our derivation for the upper bounds based on $C_T$ and $S_T$ is non-trivial in comparison with (Zhang et al., 2017). In addition to considering mirror descent instead of gradient descent, we do not require the cost functions to be Lipschitz continuous. Thus, our regret bound analysis is substantially different from that of (Zhang et al., 2017).
>
> Q3 - The discussion at the top of page 3 ("$G_T$ can be smaller than both $C_T$ and $S_T$, especially when the cost functions fluctuate drastically over time") is confusing: if cost functions fluctuate drastically over time, shouldn't the gradients experienced always be large? Seems like the opposite statement is the usual motivation for path length-based regret: "$C_T$ and $S_T$ can be smaller than $G_T$, if the functions fluctuate a lot but it's not worst-case because the minimizers are stable". A clarification, example, or mathematical statement would be helpful.
>
> A3 - Please note that, in the quoted statement, the fluctuation of the cost functions is $\text{over time}$, instead of over the $\text{domain}$ of the cost functions. Therefore, such fluctuation is not directly related to the gradient, which is defined over the domain of the cost functions. In particular, consider the case where the cost functions switch between the two specific functions $h_1(\cdot)$ and $h_2(\cdot)$. In this case, the minimizers switch between two minimizers corresponding to $h_1(\cdot)$ and $h_2(\cdot)$. The path length $C_T$ and squared path length $S_T$ reflect the cumulative distance of successive minimizers $||x_{t-1}^* - x_{t}^*||$, which can be large when minimizers jump between two points. Now let us see what happens to $G_T$ in this scenario. By smoothness (and assuming $\nabla f_t(x_t^*) = 0$) we have $|| \nabla f_t(x_t)|| \leq L* ||x_t - x_t^*||$. Therefore, $G_T$ depends on the distance $||x_t - x_t^*||$. Furthermore, in this case OMMD produces a sequence of decisions that lie between the minimizers of $h_1(\cdot)$ and $h_2(\cdot)$. Hence, $G_T$ can be smaller than $C_T$ and $S_T$ when the functions fluctuate drastically over time.
>
> Q4 - Remark 4 is also confusing: if f needs to be strongly convex with a constant lambda, then its gradient dual norm can't shrink uniformly. Could the authors clarify?
>
> A4 - First, we apologize that there is typo in Remark 4, where $||\nabla f_t (x)||$ should be $||\nabla f_t (x_t)||$. Then, the following explains how the gradients can shrink. Consider the conventional definition of strong convexity, .i.e., $f(y) \geq f(x) + \langle \nabla f(x), y - x \rangle + \frac{\lambda}{2}||y-x||^2$. By taking gradient with respect to $y$, we obtain $||\nabla f(y) - \nabla f(x)|| \geq \lambda ||y - x||$. By setting $x = x^*$ as the minimizer of the function $f(\cdot)$ (and assuming $\nabla f(x^*) = 0$), we have $||\nabla f(y)|| \geq \lambda ||y - x^*||$. Therefore, it is possible that the gradient shrinks as $y$ approaches the minimizer $x^*$.

---

> > ### Author Response · Authors · 2020-11-17
> > **Response to Reviewer 2 (2/2)**
> >
> > Q5 - The fact that the Lipschitzness of the loss functions isn't assumed is also not necessarily a feature. At points where $f_t$ is highly non-Lipschitz, the gradient is large, and thus $G_T$ (the upper bound claimed to be the novelty) is large. While it's true that the regularizer r can be chosen more freely than the losses, a poor choice leads to large $L_r$, so it's not clear what claim that discussion (at the end of 2.1) is making.
> >
> > A5 - Lipschitz continuity requires the dual norm of the gradient to be bounded for every points in the domain. Without the Lipschitz continuity assumption, while the dual norm of the gradient of some functions may not be bounded for the entire domain, it is possible that they have small gradients for many feasible points. Now, the regret bound based on $G_T$ depends on the gradients evaluated at the learner's decision. The gradient at these points is not expected to be large since they are close to the minimizers as Lemma 1 suggests. Therefore, it is possible for $G_T$ to be small even when the cost functions are not Lipschitz continuous.
> > Prior works commonly assumed Lipschitz continuity of the cost functions to derive their performance bounds. There is a notable weakness in such bounds. Since the sequence of cost functions are revealed to the learner, the learner has no control over it. If these cost functions happen to not meet the Lipschitz condition, any analysis that requires this condition becomes inapplicable. In this work, we do not require the cost functions to be Lipschitz continuous . Instead, we have moved the Lipschitz continuity condition from the cost functions to the Bregman divergence to broaden the application of our work ($K$ is the Lipschitz factor of Bregman divergence, and $L_r$ is the smoothness factor associated with the regularization function $r(\cdot)$). The main benefit of this is that the regularization function and the corresponding Bregman divergence $\text{is within the control of the learner}$. The learner can carefully design this regularization function to satisfy the Lipschitz continuity of the associated Bregman divergence with a small factor. Therefore, we believe that the proposed solution to remove Lipschitz continuity of the cost functions is an significant improvement over the current state of the art. In the revision, we have added remarks to clarify this point.
> > Finally, we would like to re-iterate that the novelty of our work is not limited to the analysis of the bound based on $G_T$, as our analysis of the bounds based on $C_T$ and $S_T$ without Lipschitz continuity is also new.

---

### Official Review · AnonReviewer1 · 2020-11-03
**Questions on the significance of the result.**

**Rating:** 4
**Confidence:** 4

**Review:**

For strongly convex & smooth functions, this work considers regret against the best sequence of points in hindsight, and gives an algorithm that achieves regret that is linear in path length and square path length while making multiple gradient queries every round.

Comments:
+ While discussing related work, the paper conflates two notions of "dynamic regret" which are quite distinct both in the nature of guarantee and techniques needed to achieve them. The first is the one paper explains -- regret against the sequence x_t^* typically scaling with the path length of x_t^*, where x_t^* are minimizers. The second is regret against ANY arbitrary sequence z_t^* with regret scaling as path length of z_t^* (again, z_t^*'s are arbitrary). A guarantee of the second kind is strictly stronger in that it implies the first (and not vice versa). Indeed, Zinkevich and Jadbabaie et al consider the second notion. Hence, the picture the paper paints of the state of prior work is not entirely accurate.
+ Now, let us just focus on the first objective as this paper does. Here, a very simple strategy -- choosing x_t = x_{t-1}^* to be the minimizer of the last function -- works. Indeed, this immediately gives regret bounds of L (Lipschitz constant) * path length, and (smoothness) beta * squared path length. Obtaining these with single step of GD offer a challenge, but allowed multiple gradient calls, one can use them to compute x_{t-1}^* offline. In this view, I don't find the results compelling.
+ The work mentions that previous works required a Lipschitz constant bound. In presence of smoothness, isn't a Lipschitz constant bound of beta * Diameter implied?
+ The first experiment seems to constrain iterates to positive orthant (for regression) via update 6. Is there a reason for this choice?

---

> ### Author Response · Authors · 2020-11-17
> **Response to Reviewer 1 (1/2)**
>
> Q1 -  While discussing related work, the paper conflates two notions of "dynamic regret" which are quite distinct both in the nature of guarantee and techniques needed to achieve them. The first is the one paper explains -- regret against the sequence $x_t^*$ typically scaling with the path length of $x_t^*$, where $x_t^*$ are minimizers. The second is regret against ANY arbitrary sequence $z_t^*$ with regret scaling as path length of $z_t^*$ (again, $z_t^*$'s are arbitrary). A guarantee of the second kind is strictly stronger in that it implies the first (and not vice versa). Indeed, Zinkevich and Jadbabaie et al. consider the second notion. Hence, the picture the paper paints of the state of prior work is not entirely accurate.
>
> A1 - We thank the reviewer for his/her helpful comments. We agree that a more general definition of dynamic regret was introduced in (Zinkevich, 2003), exactly as the reviewer has stated. However, similar to most prior works, in this work we focus on a more typical definition of dynamic regret that uses the sequence of minimizers as comparison target, to compete against the best possible solution sequence that obtains the smallest cumulative cost. We note that the regret bounds developed in (Zinkevich, 2003) and (Jadbabaie et al., 2015), for the more general definition, also hold for this specific definition (since they hold for any arbitrary $z_t^*$, including the case $z_t^* = x_t^*$). In the revision, we have added remarks to clarify that the regret bounds in (Zinkevich, 2003) and (Jadbabaie et al., 2015) apply to more general comparison targets.
>
> Q2 -  Now, let us just focus on the first objective as this paper does. Here, a very simple strategy -- choosing $x_t = x_{t-1}^*$ to be the minimizer of the last function -- works. Indeed, this immediately gives regret bounds of L (Lipschitz constant) * path length, and (smoothness) beta * squared path length. Obtaining these with single step of GD offer a challenge, but allowed multiple gradient calls, one can use them to compute $x_{t-1}^*$ offline. In this view, I don't find the results compelling.
>
> A2 -First, we would like to re-iterate an important point that in our work, the common condition of Lipschitz continuity of the cost functions is NOT assumed. Therefore, one cannot obtain the regret bound suggested by the reviewer by choosing $x_t = x_{t-1}^*$. Instead, choosing $x_t = x_{t-1}^*$ results in a regret bound of the form smoothness factor*squared path length.
>
> More importantly, however, the optimization approach of choosing $x_t=x_{t-1}^*$ and its regret bound are not very useful, since the computation of the exact minimizer $x_{t-1}^*$ requires many iterations.  In particular, let us consider offline gradient descent on a $\lambda-$strongly convex and $L-$smooth cost function with minimizer $x^*$. At the $k+1-$th step of gradient descent we have $||x_{k+1} - x^*||^2 \leq \exp( - \frac{k \lambda}{L}) ||x_1 - x^*||$ (Theorem 3.10 in (Bubec, 2014)). Therefore, to obtain the minimizer with a precision of $\epsilon$, one needs to perform $O(\frac{L}{\lambda} \log(\frac{R}{\epsilon}))$ steps, where $R$ represents the diameter of the decision set. Accurate computation of the minimizer requires $\epsilon$ to be small, which in turn implies a large number of steps per round. Thus, it is prohibitive to repeat this procedure in every online round. In comparison, the number of steps per round in OMMD is in  $O(\lceil \left(\frac{1}{2} + \frac{1}{2\alpha \lambda}\right) \log L_r \rceil)$, where $\alpha$ and $L_r$ represent the steps size and smoothness factor of the regularization function, respectively. The number of steps per round in OMMD to choose $x_t$ does not depend on the ratio of $\frac{R}{\epsilon}$, so it is significantly smaller than the number of steps per round to compute $x_{t-1}^*$.
>
> Q3 - The work mentions that previous works required a Lipschitz constant bound. In presence of smoothness, isn't a Lipschitz constant bound of beta * Diameter implied?
>
> A3 - The cost functions that arise in many practical applications are not Lipschitz continuous (Antonakopoulos et.~al., 2020). This has been the motivation of recent works to study the performance of optimization methods beyond this classical assumption (Bauschke et al., 2017; Lu et al., 2018; Antonakopoulos et al., 2020; Zhou et al., 2020). In this work, we have relaxed the Lipschitz continuity assumption and used a weaker condition of smoothness. Generally, smoothness does not imply Lipschitz continuity.  Even when the set is bounded, a Lipschitz factor of smoothness factor*diameter is usually  excessively large due the diameter term.

---

> > ### Author Response · Authors · 2020-11-17
> > **Response to Reviewer 1 (2/2)**
> >
> > Q4 - The first experiment seems to constrain iterates to positive orthant (for regression) via update 6. Is there a reason for this choice?
> >
> > A4 - The negative entropy regularization function $r(x) = \sum_{i = 1}^{d} x_i \log(x_i)$ is implicitly constrained by the function domain. As required by the domain of the $\log(\cdot)$, decisions generated via closed-form update in eq. (6) belong to the positive orthant.
> >
> > References cited above:
> >
> > M. Zinkevich.  Online convex programming and generalized infinitesimal gradient ascent.  InProceedings of the International Conference on Machine Learning, 2003
> >
> > A. Jadbabaie, A. Rakhlin, S. Shahrampour, and K. Sridharan. Online optimization:Competing with dynamic comparators. In Proceedings of the International Conference on Artificial Intelligence and Statistics, 2015.
> >
> > S. Bubeck. Convex optimization: Algorithms and complexity. arXiv preprintarXiv:1405.4980, 2014
> >
> > H. H. Bauschke, J. Bolte, and M. Teboulle. A descent lemma beyond lipschitz gradient continuity: first-order methods revisited and applications. Mathematics of Operations Research, 42(2):330–306 348, 2017
> >
> > H. Lu, R. M. Freund, and Y. Nesterov. Relatively smooth convex optimization by first-order methods, and applications. SIAM Journal on Optimization, 28(1):333–354, 2018
> >
> > K. Antonakopoulos, E. V. Belmega, and P. Mertikopoulos. Online and stochastic optimization beyond lipschitz continuity: A riemannian approach. International Conference on Learning Representations, 2020
> >
> > Zhou, et al. Regret bounds without Lipschitz continuity: online learning with relative-Lipschitz losses. Advances in Neural Information Processing Systems, 2020

---

### Decision · Program_Chairs · 2021-01-07
**Final Decision**

**Decision:**

Reject

**Comment:**

All the reviewers questioned the significance of the result, in the sense that the qualitatively it is not clear how much of an improvement it is to replace "min(S_T,C_T) with Lipschitz assumption" by "min(S_T,C_T,G_T)". The authors' response on this point did not convince the reviewers. If the authors were to resubmit this work to a future conference, we encourage them to significantly expand on this point.